# Coumarin Derivatives as New Toxic Compounds to Selected K12, R1–R4 *E. coli* Strains

**DOI:** 10.3390/ma13112499

**Published:** 2020-05-30

**Authors:** Paweł Kowalczyk, Arleta Madej, Daniel Paprocki, Mateusz Szymczak, Ryszard Ostaszewski

**Affiliations:** 1Department of Animal Nutrition, The Kielanowski Institute of Animal Physiology and Nutrition, Polish Academy of Sciences, Instytucka 3, 05-110 Jabłonna, Poland; 2Institute of Organic Chemistry, Polish Academy of Sciences, Kasprzaka 44/52, 01-224 Warsaw, Poland; arleta.madej@icho.edu.pl (A.M.); dpaprockizolw@gmail.com (D.P.); ryszard.ostaszewski@icho.edu.pl (R.O.); 3Department of Molecular Virology, Institute of Microbiology, Faculty of Biology, University of Warsaw, Miecznikowa 1, 02-096 Warsaw, Poland; mszymczak@biol.uw.edu.pl

**Keywords:** oxidative stress, coumarin derivatives, Fpg protein—formamidopyrimidine DNA *N*-glycosylase/AP lyase, LPS—lipopolysaccharide

## Abstract

Coumarins are natural compounds that were detected in 80 species of plants. They have numerous applications including the medical, food, tobacco, perfumery, and spirit industries. They show anti-swelling and diastolic effects. However, excess consumption of coumarins may adversely affect our health, because they are easily absorbed from the intestines into the lymph and blood, causing cirrhosis of the liver. Peptidomimetics are molecules whose structure and function are similar to those of peptides. They are an important group of compounds with biological, microbiological, anti-inflammatory, and anti-cancer properties. Therefore, studies on new peptidomimetics, which load the effect of native peptides, whose half-life in the body is much longer due to structural modifications, are extremely important. A preliminary study of coumarin analogues and its derivatives as new potential antimicrobial drugs containing carboxylic acid or ester was performed to determine their basic structure related to their biological features against various types of Gram-stained bacteria by lipopolysaccharide (LPS). We hypothesized that the toxicity (antibacterial activity) of coumarin derivatives is dependent on the of LPS in bacteria and nature and position of the substituent which may be carboxylic acid, hydroxyl groups, or esters. In order to verify this hypothesis, we used K12 (smooth) and R1–R4 (rough) *Escherichia coli* strains which are characterized by differences in the type of LPS, especially in the O-antigen region, the outermost LPS layer. In our work, we synthesized 17 peptidomimetics containing a coumarin scaffold and checked their influence on K12 and R1–R4 *E. coli* strains possessing smooth and rough LPS. We also measured the damage of plasmid DNA caused by target compounds. The results of our studies clearly support the conclusion that coumarin peptidomimetics are antagonistic compounds to many of the currently used antibiotics. The high biological activity of the selected coumarin peptidomimetic was associated with identification of the so-called magic methyl groups, which substantially change the biochemical properties of target compounds. Investigating the effects of these compounds is particularly important in the era of increasingly common resistance in bacteria.

## 1. Introduction

The Passerini reaction can be used for simple synthesis of libraries of compounds like coumarins and their derivatives, with can be later used for biological studies [1,2,3,4,5,6,7]. Coumarins are classified as secondary metabolites of plants, and they were isolated from more than 80 species of plants and microorganisms [8]. They were first discovered in 1820 and synthesized in 1868. These aromatic compounds are widely used in medicine; for example, coumarin derivatives have a toxic effect on cancer cells, preventing inflammation and working against blood clots [9,10,11,12,13]. Pure coumarins are much more reactive than commonly used antibiotics [12,13]. For example, drugs such as captopril, acenocoumarol, and warfarin are vitamin K antagonists widely used as anticoagulants [14]. Coumarin derivatives are used in the cosmetics, spirit, tobacco, and natural medicine industries [15]. Anti-inflammatory and antiviral properties of coumarin derivatives were also evaluated on *Mycobacterium tuberculosis* [16,17,18]. Currently, there are insufficient data in the professional scientific literature on the cytotoxic properties of simple synthetic coumarin derivatives performed on *Escherichia coli* R1–R4 model bacterial cells. Scientists only focused on the action of antibiotics produced by *Streptomyces* species such as clorobiocin or novobiocin, which contain coumarin in their composition and may be potential compounds that inhibit bacterial gyrase and topoisomerases [18]. Coumarin and its compounds were recently identified as potential substances that inhibit the activity of proteases and integrases in the study of the antimicrobial profile [19,20].

Investigating newly synthesized coumarin derivatives and their potentially toxic effect on bacterial cells may be an alternative to many new cases of drug resistance (including antibiotics) as effective antimicrobials against bacterial clinical pathogens.

Based on these studies, attempts were made to determine the toxic activity of compounds having a coumarin scaffold. Carboxylic acid or esters were used as substituents on the coumarin ring. Attachment of these substituents was important for demonstrating strong toxic effects on all bacterial cells [19,20,21].

The insertion of a carboxylic acid or phenolic hydroxyl group in coumarin derivatives is important in achieving a toxic effect on the analyzed *E. coli* bacteria, which means that these compounds have broad antibacterial potential. Another possibility is the use of coumarins as fluorescent markers for the determination of living cells and enzyme activities, as well as fluorescent sensors used to study pH changes, for the detection of hydrogen peroxide, nitroxides, or nitrogen oxides [22]. For example, coumaric acid is an organic chemical compound from the group of phenolic acids, a derivative of cinnamic acid. Due to the location of the hydroxyl group, three isomers can be distinguished: *ortho*, *meta*, and *para*. The *para* isomer is the most common isomer in nature. 

Coumaric acid is one of the main building blocks of lignocellulose and is found in many crop species with antioxidant activity [1].

It potentially reduces the risk of stomach cancer by reducing the level of nitrosamines [2]. The proposed peptidomimetics with coumarin scaffolds usually contain an alkylating ring that can strongly interact with the bacterial components of lipid membranes and hydrophobic parts of synthetic compounds. 

We hypothesize that the interaction of coumarin derivatives may also depend on the length of lipopolysaccharide (LPS) in the bacterial cells. 

To confirm this hypothesis, in the present work, we applied the base control strain K-12 (without LPS) and *Escherichia coli* R1–R4 model strains (having LPS of different length at the location of the O antigen in their outer layer). Lipopolysaccharide (LPS) is an amphiphilic molecule, surface antigen, and virulence factor which constitutes up to 80% of the outer membrane part of bacteria not stained by the Gram method [23]. LPS also plays a key role in the interaction of the host [24] with its environment [25]. *E. coli* LPS contains, in its composition, lipid A, which has hydrophilic properties, and core polysaccharides and oligosaccharides, which together form the O-antigen [26,27]. Based on the literature, we predicted that the interaction of coumarin derivatives will protect against oxidative stress processes and DNA damage, as observed after interaction with ionic liquids [28]. In our previous studies, we showed that a series of α-acyloxycarboxamides can easily be obtained via the Passerini reaction, which can be obtained in environmentally friendly conditions [29,30,31].

## 2. Results and Discussion

### 2.1. Chemistry

For biological studies, we chose compounds with a coumarin scaffold: coumarin-3-carboxylic acid (1) and ethyl 3-coumarincarboxylate (2). For comparison, we prepared peptidomimetics without a coumarin scaffold, i.e., compound 3, via the Passerini reaction involving an acetic acid, a dodecanal, and a *p*-methoxybenzyl isocyanide. The reaction was carried out in dichloromethane, and ester 3 was obtained with 59% yield. Biological studies were also carried out for commercially available 1,3-dioxane-2-one (17). The structure of compounds 1, 2, 3, and 17 is presented in the Figure 1.

Peptidomimetics containing a coumarin scaffold were synthetized via the Passerini reaction. Substrates were stirred in dichloromethane, and compounds 4–16 were isolated chromatographically in yields summarized in Table 1.

### 2.2. Antimicrobial Properties of the Used Compounds

Used peptidomimetics has a different effect on the minimum inhibitory concentration (MIC) which is associated with different lengths of the alkyl chain (Figure 1).

MIC values for individual strains of *E. coli* R1–R4 and K12 were visible on all analyzed microplates after the addition of the indicator microbial growth (resazurin). On the first plate, strains K12, R1, R2, and R3 were sequentially added together with another three compounds, i.e., coumarinacid 1, ethyl ester of coumarin acid 2, and peptidomimetic 3 without a coumarin group. Here, a color change was observed already at a dilution of 10^−3^ and a MIC value of 0.02 mg∙mL^−1^. On the second plate, strains K12, R1, R2, and R3 were sequentially added together with another three peptidomimetic compounds marked 11, 12, and 13. Here, a color change was observed already at a dilution of 10^−4^ and a MIC value of 0.002 mg∙mL^−1^. On the third plate, strains K12, R1, R2, and R3 were sequentially added together with the next three peptidomimetic compounds designated as 5, 15, and 6. Here, a color change was observed for compound 5 only at a dilution of 10^−5^ corresponding to the MIC value of 0.002 mg∙mL^−1^. However, changes in the color in the wells after using peptidomimetics 5, 6, and 15 were visible at a dilution of 10^−3^, corresponding to an MIC value equal to 0.02 mg∙mL^−1^. On the fourth plate, strains K12, R1, R2, and R3 were sequentially added together with three peptidomimetics marked as 9, 4, and 3. Here, a color change was observed in strain K12 for all three analyzed compounds at a dilution of 10^−3^ corresponding to an MIC equal to 0.02 mg∙mL^−1^. In contrast, for the other R1–R3 cavities, color changes in the wells after using peptidomimetics 9, 4, and 3 were visible at a dilution of 10^−4^, corresponding to an MIC value of 0.002 mg∙mL^−1^. On the fifth plate, strains K12, R1, R2, and R3 were sequentially added together with the three peptidomimetics designated as 7 and 16. Here, in all strains, a color change in the wells was observed for the analyzed compounds even at a dilution of 10 corresponding, to an MIC value of 0.002 mg∙mL^−1^. On the sixth plate, strains K12, R1, R2, and R3 were sequentially added together with three peptidomimetic compounds marked 2, 14, and 17. Here, in all strains, a color change in the wells was observed for the analyzed compounds at a dilution of 10^−2^, corresponding to an MIC equal to 0.2 mg∙mL^−1^. On the seventh and eighth plates, only the R4 strain was applied together with further peptidomimetic compounds (coumarin derivatives) determined from 1 to 17. In all analyzed wells with the R4 strain, a color change was observed for the tested compounds from 1 to 12 already at a dilution of 10^−4^, corresponding to an MIC value of 0.002 mg∙mL^−1^. In the eighth plate, where compounds 13 to 17 were analyzed, the color change was visible already at a dilution of 10^−3^, corresponding to an MIC value equal to 0.02 mg∙mL^−1^. Mutual interactions between the analyzed compounds and strains on the basis of plaque analysis are presented in Figure 2.

In the experiments, MIC and minimum bactericidal concentration (MBC) values were determined for all analyzed coumarin derivatives (Figure 2, Figure 3 and Figure 4). Differences in MIC values for *E. coli* strains (differing in LPS length) and the K12 control strain were observed in all analyzed microtiter plates after the addition of resazurin, which indicated microbial growth (Appendix A).

Increasing MIC values were observed for all 17 analyzed compounds. Bacterial strains R2 and R4 displayed different but higher sensitivity to peptidomimetics, but R4 strains were highest. The R4 strain was probably the most sensitive, compared to other strains, due to the length of the lipopolysaccharide chain. In all analyzed cases, MBC values were about 20 to 75 times higher than the MIC (Figure 4).

The *E. coli* R4 strain did not show (despite the largest LPS length) a high resistance to toxic concentrations of the test compounds, as shown by the MBC/MIC ratio (Figure 4). The modification of functional groups in coumarin derivatives probably changes the MBC/MIC ratio and strongly depends on the used K12 strain versus R strains (Figure 4).

The obtained MIC and MBC values between individual strains after treatment with the analyzed compounds were statistically significant, as outlined in Table 2.

### 2.3. Modification of Plasmid DNA Isolated from E. coli R1–R4 Strains with Tested Coumarin Derivatives

MIC values revealed that the toxicity of test compounds should increase with increasing alkyl chain length. We expected a similar effect in experiments with isolated plasmid DNA and modified coumarin derivatives, where plasmid damage should be most apparent.

On the basis of MIC values, two strains (K12 and R4) were selected for further DNA analysis. The 17 compounds did not interfere with the DNA structure even when digesting the Fpg (formamidopyrimidine DNA *N*-glycosylase/AP lyase) protein in vitro during a 24 h experiment.

The Fpg protein digestion of plasmids isolated from both control and cultures treated with peptidomimetics, regardless of the length of the alkyl chain, clearly showed no visible damage in the mutual changes of covalently closed circle (ccc), linear, and open circle (oc) forms, as well as in fuzzy bands known as the “smear” (Figure 5). In the case of plasmids from strains K12 and R4, three traditional forms were observed: very poor form oc, linear, and ccc. We did not observe important differences between the control and plasmids modified by peptidomimetics in the electrophoretic images (Figure 5).

The results of plasmid DNA modified by coumarin derivatives presented in Figure 5 and Figure 6 showed that all analyzed peptidomimetics with different alkyl chain length and substituents containing a phenolic hydroxyl group or carboxylic acid did not change the topological forms of plasmids, even after digestion with the Fpg protein.

In plasmid DNA isolated from both K12 and R4 strains, mainly the oc and linear forms were observed. Only about 3% oxidative damage after Fpg digestion was identified, which may indicate that coumarin derivatives do not damage plasmid DNA, and that the interaction with genetic material occurs via development of oxidative stress in living cells. Moreover, the microbial toxicity of the targeted compounds may be affected by the composition of lipopolysaccharide (LPS) in bacteria.

Our observations show that the length of the alkyl chain of peptidomimetics may determine toxicity for specific *E. coli* R-type strains.

It was observed that the structural properties of individual compounds had an effect on antimicrobial activity in all analyzed bacterial strains, as can be seen from the MIC and MBC values.

The obtained MIC and MBC values were visible generally at the dilutions of 10^−3^ or 10^−4^ mL and concentrations of 0.02 or 0.002 mg∙mL^−1^ in all analyzed strains. 

The antimicrobial efficacy of the long-chain compounds used was, therefore, higher (based on MIC and MBC values) in the aggregated state in compounds 7, 8, 10, 14, and 17 with respect to control relationships and compounds 1 and 2 (lack of the microbial activity). This means that the structure of the compound is of fundamental importance for its activity. 

Our results indicate that peptidomimetics containing, in their structure, carboxylic acid groups or esters may be toxic to bacterial cells, while peptidomimetics containing, in their structure, a phenolic hydroxyl group show significantly lower toxicity in relation to *E. coli* bacterial cells. It is known from the literature data that *E. coli* can induce gastrointestinal diseases, as well as intestinal and stomach cancers [31,32].

In relation to the lack of differences between compounds 4, 5, and 6 (Figure 2), in compound 7, the methyl group in the 6-position of coumarin significantly lowered the MIC, especially with respect to the similar unsubstituted analogue 4. In compound 16, an isopropyl group in the peptidomimetic structure greatly increased the value of the MIC. The potential toxicity of peptidomimetics to bacterial cells is indicated by the fact that all used strains were sensitive to tested peptidomimetics, while R2 and R4 were the most sensitive, compared to K12, R1, and R2 [33]. 

For short-chain alkyl molecules (Table 1), MBC values were higher than the MIC for all strains [28]. The tested compounds without a long alkyl chain were more effective than molecules with a long alkyl chain for the K12 and R1–R3 strains. The effect was, therefore, higher in the aggregated state after interaction with ionic liquids [28]. A similar effect was shown with the quaternary ammonium surfactants [28,29,34]. A further increase due to, e.g., surfactant concentration, may lead to the release of membrane components into the cell environment and block the transmission of electrons [28,29]. 

Changes in the chemical composition of the membrane and loosening of its structure can lead to its disintegration [34]. The results of our research suggest that further toxicity studies based on the analyzed compounds over other strains of K12, R1–R4 bacteria are necessary to determine potential mechanisms for the disintegration of their cell membrane.

The toxicity analysis of the peptidomimetics used is strongly correlated with the length of lipopolysaccharide in the individual types of R1–R4 bacterial concepts used. The results of our research indicate that, in all peptidomimetics used, strains R2 and R4 were most sensitive to their toxic effects compared to strains R1 and R3.

The results were estimated based on a comparison of samples digested and non-digested with Fpg protein. The percentage cleavage was determined based on the change in forms resulting from the enzymatic activity of the digested sample relative to the DNA-binding Fpg protein by all tested peptidomimetics. From literature data, it is known that the Fpg enzyme (as a two-functional glycosylase) removes a broad spectrum of oxidized and alkylated bases from DNA [35] such as 7,8-dihydro-8-oxoguanine (8-oxoguanine), 8-oxoadenine, substituted and unsubstituted purines introduced into DNA by reactive oxygen species (e.g., FapyA, FapyG) [35], anticancer drugs, and chemical carcinogens, (e.g., Fapy-7-aminoethylG, aflatoxin, B_1_-fapy-guanine, 5-hydroxy-cytosine, 5-hydroxy-uracil, Fapy-7EtG, Fapy-7MeG) [36,37,38,39,40,41].

No damage after digestion with Fpg protein, as the most sensitive marker of oxidative stress recognizing the oxidized bases of DNA in the cell, suggests that the remaining repair enzymes from the entire cascade of the DNA infusion system would not show activity; hence, they were not tested here. The degree of DNA damage recognized by the Fpg protein above 3% is an important indicator of modified and oxidized guanines occurring as 8-oxoguanine, fapy-adenine (FapyA), and fapy-guanine (FapyG). The Fpg enzyme probably does not recognize modifications after the reaction of peptidomimetics with bacterial plasmid DNA, as we did not observe any disrupted DNA between oc, linear, and ccc forms digested by Fpg.

The results suggest that the tested peptidomimetics, except for compound 17 (see Table 1), did not introduce oxidative DNA modifications leading to the formation of oxidized substrates recognized by the Fpg protein. The rational design of coumarin peptidomimetic derivatives marked 7, 8, 10, and 14 (see Table 1) as new drug substitutes (e.g., antibiotics, which have a similar structure as peptidomimetics used in the experiment) may be associated with increased resistance to specific enzymes/proteases, as well as the high cell membrane permeability in bacteria and systemic cellular barriers in higher organisms. Interesting properties of MIC and MBC were observed for compound 7, indicating its super selectivity to the membranes of Gram-negative bacterial cells. The obtained compounds may stimulate a given physiological process to a greater extent than the native peptide, while those with antagonistic functions may be reversed. Target systems for the action of peptidomimetics may be the neuro-endocrine, digestive, excretory, and reproductive systems, as well as the heart or muscles. 

It is known from the literature that chemotherapy with antibiotics such as amoxicillin, metronidazole, clarithromycin, and even bismuth compounds [42] used on *E. coli* bacteria led to an increase in their resistance. Therefore, the search for new specific compounds that, in their structure and function, would be analogues of antibiotics but more toxic in their action on bacterial cells seems to be justified.

The functioning of these systems and organs in insects is regulated by neuropeptides and peptide hormones. Based on our knowledge, we can precisely design mimetics with high selectivity of tissue function. Currently, one of the most important research tasks is a broader understanding of the mechanisms of interaction of these compounds on various structures at both the cellular and the molecular level. Particular attention should be paid to specific receptors in the nervous system and peripheral tissues, which are important in the design of highly effective and selective mimetics. There is still a need to obtain more fundamental knowledge about the action of already synthesized and newly designed peptidomimetics in vivo using high-throughput screening bioassays in bacterial systems for both types of Gram-stained bacteria. Another important task during the development of the practical use of various peptidomimetics will be to understand their physiological effects under environmental conditions. In this regard, before the commercial use of these compounds, further studies on biostability, half-life, the most preferred mode of administration, and the potency of peptidomimetics will be required.

## 3. Conclusions

The antibacterial (toxic) effect of the tested compounds was significantly different compared to the tested *E. coli* strains K12 and R1–R4. In our study, we showed that the peptidomimetics used may be toxic to *E. coli* R1–R4 strains containing different lengths of LPS, with a specificity in growth rate and metabolic activity, based on MIC and MBC tests. Experiments showed that compounds tested in vitro with isolated plasmid DNA and treated with Fpg protein did not react directly with it. No noticeable changes were found between individual forms of plasmid DNA (oc, linear, and ccc forms), which suggests that Fpg glycosylase does not recognize the oxidative or alkylating modifications directly induced by peptidomimetics that may be potential substrates for this protein. Toxic properties of coumarin derivatives damaging bacterial cells were observed after their direct entry into them. Thus, the DNA damage generated by the oxidative stress arising in the cell is strongly correlated with the type of substituent and the length of the alkyl chain. Our research confirms the hypothesis that the dependence of the structure of simple coumarins on their antibacterial activity depends on the nature and position of the substituent, which may be carboxylic acid, hydroxyl groups, or esters. 

Our results are important for research into the mechanism of toxic effects of new drugs (peptidomimetics) based on coumarin derivatives that can damage the cell membrane of bacteria by changing their surface charge, which may play an important role in reducing antibiotic resistance. This is particularly relevant for substituents at the 6-position of coumarin (R1 = methyl group in compound 12) and the R2 alkyl group (R2 = *i*−Pr in compound 16), which may determine the biochemical changes in the activity of coumarin-derived peptidomimetics as new antibiotic antagonists, whereby a special effect was observed for compounds 7, 8, 10, and 14, which displayed the lowest MIC values and MBC/MIC ratios. Compound 7 showed super selectivity in all analyzed bacterial strains. The studies presented apply only to bacteria, and other cytotoxicity studies using different cell lines and cultures should be carried out in the future to assess the biocompatibility of the test compounds.

## 4. Experimental Section

### 4.1. Strains and Media

*E. coli* strains R1–R4 and K-12 were obtained as a kind gift from Prof. Jolanta Łukasiewicz at the Ludwik Hirszfeld Institute of Immunology and Experimental Therapy (Polish Academy of Sciences). Bacteria were grown in liquid medium or agar plates containing tryptic soy broth medium (TSB; Sigma-Aldrich, Saint Louis, MI, USA). *N*,*N*-Dimethylformamide (DMF) was obtained from Sigma Aldrich (CAS No. 68-12-2). 

### 4.2. Experimental Chemistry

NMR analyses were acquired using a Bruker 400 spectrometer (400 MHz for ^1^H-NMR, 100 MHz for ^13^C-NMR, Kennewick, WA, USA) with tetramethylsilane (TMS) used as an internal standard or the residual chloroform signal. High-resolution mass spectrometry (HRMS, Waltham, MA, USA) spectra were recorded on an Mariner (PerSeptiveBiosystems) and Synapt G2:SHD apparatus. Column chromatography was performed using Kiesegel 60 (230–400 mesh, Sigma-Aldrich, Saint Louis, MI, USA). All solvents, chemicals, and reagents were purchased from Sigma-Aldrich or Tokyo chemical industry (TCI, Montgomeryville, PA, USA) and used without further purification.

### 4.3. General Procedure for Synthesis of Compounds 3–17 

The mixture of carboxylic acid (0.5 mmol), aldehyde (0.5 mmol), and isocyanide (0.5 mmol) was stirred in dichloromethane (5 mL) for 24 h at room temperature (RT). Then, solvent was removed by distillation under reduced pressure. The crude products were purified by column chromatography on silica gel using hexane/AcOEt as the eluent. The analysis of NMR spectra of obtained compounds consisted of several signals which can be identified for several organic compounds. Only the full set of those signals is characteristic for each compound. The possible coincidence of several identical signals is meaningless for identification. The general synthetic procedure is intentionally close to the general synthetic procedure used for the synthesis of Passerini products. The standard synthesis of compounds is methodologically identical to that presented in other publications of this type. Structural formulas of coumarin derivatives are presented in Appendix A.

#### 4.3.1. 1-(4-Methoxybenzylamino)-1-Oxotridecan-2-yl Acetate 3

^1^H-NMR (400 MHz; CDCl_3_) δ 0.88(3H, t, J 7.2 Hz, CH_3_CH_2_), 1.21–1.35 (18H, br m, 9× CH_2_), 1.80–1.91 (2H, m, CH_2_CH), 2.11 (3H, s, CH_3_CO), 3.90 (3H, s, CH_3_O), 4.34–4.45 (2H, m, CH_2_N), 5.16–5.20 (1H, m, CH), 6.23 (1H, t, J 5.2 Hz, NH), 6.21–6.24 (2H, m, Ph), 7.18–7.20 (2H, m, Ph); ^13^C-NMR (100 MHz; CDCl_3_) δ 14.1, 21.0, 22.7, 24.8, 29.2, 29.3, 29.4, 29.5, 29.6, 31.9, 31.9, 42.7, 55.3, 74.2, 114.1, 129.0, 129.9, 159.1, 169.6, 169.7; HRMS calculated for C_23_H_37_NO_4_Na [M + Na]^+^: 414.2620, found: 414.2614.

#### 4.3.2. 1-(4-Methoxybenzylamino)-1-Oxotridecan-2-yl-3-Coumarincarboxylate 4

^1^H-NMR (400 MHz; CDCl_3_) δ 0.86 (3H, t, J 6.9 Hz, CH_2_CH_3_), 1.16–1.45 (18H, m, 9× CH_2_), 1.89–2.08 (2H, m, CHCH_2_), 3.77 (3H, s, OCH_3_), 4.38–4.51 (2H, m, NCH_2_), 5.54 (1H, dd, J 6.7, 4.3 Hz, CH_2_CH), 6.82–6.87 (2H, m, Ph), 7.14–7.30 (1H, m, Ph), 7.34–7.41 (3H, m, Ph), 7.62–7.71 (2H, m, Ph), 8.20–8.27 (1H, m, NH), 8.56 (1H, s, C=CH); ^13^C-NMR (100 MHz; CDCl_3_) δ 14.1, 22.6, 29.2, 29.3, 29.4, 29.5, 29.6, 29.7, 31.9, 42.6, 55.2, 75.6, 114.0, 116.9, 117.8, 118.1, 125.3, 129.1, 129.7, 130.5, 134.8, 149.8, 155.1, 157.9, 158.8, 162.6, 169.5; HRMS calculated for C_31_H_39_NO_6_Na [M + Na]^+^: 544.2675, found: 544.2669.

#### 4.3.3. 1-(4-Methoxybenzylamino)-1-Oxotridecan-2-yl-3-(6-Nitrocoumarin) Carboxylate 5

^1^H-NMR (400 MHz; CDCl_3_) δ 0.86 (3H, t, J 6.9 Hz, CH_2_CH_3_), 1.15–1.37 (18H, m, 9× CH_2_), 1.90–2.09 (2H, m, CHCH_2_), 3.78 (3H, s, OCH_3_), 4.37–4.50 (2H, m, NCH_2_), 5.53 (1H, dd, J 6.7, 4.3 Hz, CH_2_CH), 6.82–6.87 (2H, m, Ph), 7.21–7.27 (2H, m, Ph), 7.50–7.54 (1H, m, Ph), 7.88 (1H, t, J 5.7 Hz, NH), 8.50–8.64 (3H, m, Ph+C=CH); ^13^C-NMR (100 MHz; CDCl_3_) δ 14.1, 22.6, 29.2, 29.3, 29.4, 29.5, 29.6, 29.7, 31.9, 42.7, 55.2, 76.1, 114.0, 117.7, 118.2, 120.3, 125.4, 129.0, 129.1, 130.2, 144.5, 148.3, 156.1, 158.1, 158.9, 161.5, 169.1; HRMS calculated for C_31_H_38_N_2_O_8_Na [M + Na]^+^: 589.2526, found: 589.2527.

#### 4.3.4. 1-(4-Methoxybenzylamino)-1-Oxotridecan-2-yl-3-(6 Methoxycoumarin) Carboxylate, 6

^1^H-NMR (400 MHz; CDCl_3_) δ 0.86 (3H, t, J 6.8 Hz, CH_2_CH_3_), 1.15–1.43 (18H, m, 9× CH_2_), 1.89–2.09 (2H, m, CHCH_2_), 3.77 (3H, s, OCH_3_), 3.86 (3H, s, OCH_3_), 4.37–4.51 (2H, m, NCH_2_), 5.53 (1H, dd, J 6.8, 4.2 Hz, CH_2_CH), 6.82–6.87 (2H, m, Ph), 7.00–7.04 (1H, m, Ph), 7.23–7.34 (4H, m, Ph), 8.29 (1H, t, J 5.8 Hz, NH), 8.50 (1H, s, C=CH); ^13^C-NMR (100 MHz; CDCl_3_) δ 14.1, 22.6, 29.2, 29.3, 29.4, 29.5, 29.6, 31.9, 42.5, 55.2, 55.9, 75.6, 110.7, 113.9, 118.0, 118.1, 123.3, 129.1, 130.6, 149.6, 149.7, 156.6, 158.1, 158.8, 162.7, 169.5; HRMS calculated for C_32_H_41_NO_7_Na [M + Na]^+^: 574.2781, found: 574.2788.

#### 4.3.5. 1-(4-Methoxybenzylamino)-1-Oxotridecan-2-yl-3-(6-Methylcoumarin) Carboxylate 7

^1^H-NMR (400 MHz; CDCl_3_) δ 0.86 (3H, t, J 6.8 Hz, CH_2_CH_3_), 1.14–1.45 (18H, m, 9× CH_2_), 1.90–2.09 (2H, m, CHCH_2_), 2.43 (3H, s, PhCH_3_), 3.77 (3H, s, OCH_3_), 4.37–4.52 (2H, m, NCH_2_), 5.46–5.57 (1H, m, CH_2_CH), 6.81–6.86 (2H, m, Ph), 7.23–7.29 (3H, m, Ph), 7.39–7.41 (1H, m, Ph), 7.45–7.50 (1H, m, Ph), 8.29 (1H, t, J 5.7 Hz, NH), 8.50 (1H, s, C=CH); ^13^C-NMR (100 MHz; CDCl_3_) δ 14.1, 20.7, 22.6, 29.3, 29.4, 29.5, 29.6, 29., 31.9, 42.6, 55.2, 75.5, 1143.0, 116.6, 117.6, 117.9, 129.1, 129.3, 130.5, 135.2, 136.0, 149.8, 153.3, 158.1, 158.8, 162.8, 169.5; HRMS calculated for C_32_H_41_NO_6_Na [M + Na]^+^: 558.2832, found: 558.2834.

#### 4.3.6. 1-(Benzylamino)-1-Oxotridecan-2-yl-3-Coumarincarboxylate 8 

^1^H-NMR (400 MHz; CDCl_3_) δ 0.86 (3H, t, J 6.8 Hz, CH_2_CH_3_), 1.17–1.40 (18H, m, 9× CH_2_), 1.95–2.10 (2H, m, CHCH_2_), 4.45–4.59 (2H, m, NCH_2_), 5.56 (1H, dd, J 6.7, 4.3 Hz, CH_2_CH), 7.21–7.40 (7H, m, Ph), 7.63–7.71 (2H, m, Ph), 8.32 (1H, s br, NH), 8.57 (1H, s, C=CH); ^13^C-NMR (100 MHz; CDCl_3_) δ 14.1, 22.6, 29.2, 29.3, 29.4, 29.5, 29.6, 31.9, 43.1, 75.6, 116.9, 117.8, 118.1, 125.3, 127.23, 127.8, 128.5, 129.7, 134.9, 138.3, 149.9, 155.1, 157.9, 162.6, 169.6; HRMS calculated for C_30_H_37_NO_5_Na [M + Na]^+^: 514.2569, found: 514.2559.

#### 4.3.7. 1-(*t*-Butylamino)-1-Oxotridecan-2-yl-3-Coumarincarboxylate 9

^1^H-NMR (400 MHz; CDCl_3_) δ 0.86 (3H, t, J 6.9 Hz, CH_2_CH_3_), 1.19–1.45 (27H, m, 9× CH_2_ + C(CH_3_)_3_), 1.90–2.01 (2H, m, CHCH_2_), 5.40 (1H, dd, J 6.0, 4.6 Hz, CH_2_CH), 7.35–7.42 (2H, m, Ph), 7.62–7.72 (3H, m, Ph + NH), 8.60 (1H, s, C=CH); ^13^C-NMR (100 MHz; CDCl_3_) δ 14.1, 22.6, 24.3, 28.7, 29.3, 29.4, 29.5, 29.6, 31.8, 31.9, 51.3, 75.7, 116.9, 117.9, 118.2, 125.2, 129.7, 134.8, 149.9, 155.2, 157.7, 162.5, 168.69; HRMS calculated for C_27_H_39_NO_5_Na [M + Na]^+^: 480.2726, found: 480.2721

#### 4.3.8. 1-(Cyclohexylamino)-1-Oxotridecan-2-yl-3-Coumarincarboxylate 10

^1^H-NMR (400 MHz; CDCl_3_) δ 0.86 (3H, t, J 6.9 Hz, CH_2_CH_3_), 1.15–1.44 (22H, m, 11× CH_2_), 1.56–1.65 (2H, m, CH_2_), 1.70–1.80 (2H, m, CH_2_), 1.85–2.05 (4H, m, 2× CH_2_), 3.70–3.85 (1H, m, NCH), 5.49 (1H, dd, J 6.6, 4.2 Hz, CH_2_CH), 7.36–7.43 (2H, m, Ph), 7.65–7.73 (2H, m, Ph), 7.84 (1H, d, J 8.1 Hz, NH), 8.59 (1H, s, C=CH); ^13^C-NMR (100 MHz; CDCl_3_) δ 14.1, 22.6, 24.4, 24.9, 25.5, 29.3, 29.4, 29.5, 29.6, 31.9, 48.1, 75.6, 116.9, 117.9, 118.2, 125.3, 129.7, 134.8, 149.9, 155.1, 158.0, 162.6, 168.5; HRMS calculated for C_29_H_41_NO_5_Na [M + Na]^+^: 506.2882, found: 506.2883.

#### 4.3.9. Ethyl 2-[2-(3-Coumarincarboxylate)-1-Oxotridecanamino]acetate 11

^1^H-NMR (400 MHz; CDCl_3_) δ 0.79–0.88 (3H, m, CH_2_CH_2_CH_3_), 1.10–1.45 (21H, m, 9× CH_2_ + OCH_2_CH_3_), 1.90–2.06 (2H, m, CHCH_2_), 5.07 (2H, dd, J 3.8, 1.9 Hz, NCH_2_), 4.14–4.22 (2H, m, OCH_2_), 5.53 (1H, dd, J 5.5, 3.8 Hz, CH_2_CH), 7.33–7.39 (2H, m, Ph), 7.62–7.71 (2H, m, Ph), 8.25 (1H, t, J 5.5 Hz, NH), 8.59 (1H, s, C=CH); ^13^C-NMR (100 MHz; CDCl_3_) δ 14.1, 22.6, 29.3, 29.3, 29.5, 29.6, 31.8, 41.1, 61.3, 75.4, 116.9, 117.8, 118.0, 125.2, 130.0, 134.9, 150.0, 155.16, 158.9, 162.5, 169.3, 170.3; HRMS calculated for C_27_H_37_NO_7_Na [M + Na]^+^: 510.2468, found: 510.2462.

#### 4.3.10. 1-(4-Methoxybenzylamino)-1-Oxononan-2-yl-3-Coumarincarboxylate 12

^1^H-NMR (400 MHz; CDCl_3_) δ 0.85 (3H, t, J 6.9 Hz, CH_2_CH_3_), 1.18–1.43 (10H, m, 5× CH_2_), 1.90–2.09 (2H, m, CHCH_2_), 3.78 (3H, s, OCH_3_), 4.39–4.51 (2H, m, NCH_2_), 5.54 (1H, dd, J 6.8, 4.3 Hz, CH_2_CH), 6.83–6.87 (2H, m, Ph), 7.24–7.30 (2H, m, Ph), 7.34–7.41 (2H, m, Ph), 7.62–7.72 (2H, m, Ph), 8.24 (1H, s br, NH), 8.56 (1H, s, C=CH); ^13^C-NMR (100 MHz; CDCl_3_) δ 14.0, 22.6, 24.5, 29.1, 29.2, 31.7, 32.0, 42.6, 55.2, 75.6, 114.0, 176.0, 118.1, 125.3, 129.1, 129.7, 130.5, 134.8, 149.8, 155.1, 157.9, 158.8, 162.6, 169.5; HRMS calculated for C_27_H_31_NO_6_Na [M + Na]^+^: 488.2049, found: 488.2051.

#### 4.3.11. 1-(4-Methoxybenzylamino)-1-Oxoheptan-2-yl-3-Coumarincarboxylate 13 

^1^H-NMR (400 MHz; CDCl_3_) δ 0.84 (3H, t, J 6.6 Hz, CH_2_CH_3_), 1.21–1.43 (6H, m, 3× CH_2_), 1.90–2.10 (2H, m, CHCH_2_), 3.77 (3H, s, OCH_3_), 4.38–4.52 (2H, m, NCH_2_), 5.54 (1H, dd, J 6.6, 4.3 Hz, CH_2_CH), 6.84–6.89 (2H, m, Ph), 7.24–7.30 (2H, m, Ph), 7.33–7.41 (2H, m, Ph), 7.65–7.71 (2H, m, Ph), 8.19–8.27 (1H, m, NH), 8.56 (1H, s, C=CH); ^13^C-NMR (100 MHz; CDCl_3_) δ 13.9, 22.4, 24.1, 31.4, 32.0, 42.6, 55.2, 75.6, 114.0, 116.9, 117.9, 125.3, 129.1, 129.7, 130.5, 134., 149.9, 155.1, 157.9, 158.8, 162.6, 169.5; HRMS calculated for C_25_H_27_NO_6_Na [M + Na]^+^: 460.1736, found: 460.1732.

#### 4.3.12. 1-(4-Methoxybenzylamino)-1-Oxobutan-2-yl-3-Coumarincarboxylate 14 

^1^H-NMR (400 MHz; CDCl_3_) δ 0.94 (3H, t, J 7.4 Hz, CH_2_CH_3_), 1.96–2.13 (2H, m, CHCH_2_), 3.78 (3H, s, OCH_3_), 4.38–4.52 (2H, m, NCH_2_), 5.52 (1H, dd, J 6.1, 4.5 Hz, CH_2_CH), 6.82–6.88 (2H, m, Ph), 7.14–7.29 (2H, m, Ph), 7.33–7.41 (2H, m, Ph), 7.62–7.72 (2H, m, Ph), 8.25 (1H, s br, NH), 8.57 (1H, s, C=CH); ^13^C-NMR (100 MHz; CDCl_3_) δ 8.7, 25.1, 42.6, 55.2, 76.3, 114.0, 116.9, 117.8, 118.1, 125.3, 129.7, 130.5, 134.9, 149.9, 155.1, 158.8, 162.6, 169.3; HRMS calculated for C_22_H_21_NO_6_Na [M + Na]^+^: 418.1267, found: 418.1255.

#### 4.3.13. 1-(4-Methoxybenzylamino)-1-oxo-3-Phenylpropan-2-yl-3-Coumarincarboxylate 15

^1^H-NMR (400 MHz; CDCl_3_) δ 3.20–3.29 (1H, m, CHCH_2a_), 3.39–3.46 (1H, m, CHCH_2b_), 3.77 (3H, s, OCH_3_), 4.30–4.49 (2H, m, NCH_2_), 5.74 (1H, dd, J 7.1, 4.1 Hz, CH_2_CH), 6.78–6.82 (2H, m, Ph), 7.08–7.23 (7H, m, Ph), 7.33–7.39 (2H, m, Ph), 7.58–7.71 (2H, m, Ph), 8.06 (1H, t, J 5.6 Hz, NH), 8.42 (1H, s, C=CH); ^13^C-NMR (100 MHz; CDCl3) δ 37.8, 42.6, 55.2, 75.6, 113.9, 116.9, 117.8, 125.3, 126.9, 128.3, 129.0, 129.6, 129.8, 130.2, 134.9, 135.7, 149.8, 155.1, 157.7, 158.8, 162.8, 168.6; HRMS calculated for C_27_H_23_NO_6_Na [M + Na]^+^: 480.1423, found: 480.1418.

#### 4.3.14. 1-(4-Methoxybenzylamino)-3-Methyl-1-Oxobutan-2-yl-3-Coumarincarboxylate 16

^1^H-NMR (400 MHz; CDCl_3_) δ 0.96 (6H, dd, J 6.9, 23.6 Hz, CH(CH_3_)_2_), 2.43–2.53 (1H, m, CHCH_3_), 3.75 (3H, s, OCH_3_), 4.35–4.53 (2H, m, NCH_2_), 5.49 (1H, d, J 2.2 Hz, OCH), 6.77–6.86 (2H, m, Ph), 7.19–7.29 (2H, m, Ph), 7.29–7.40 (2H, m, Ph), 7.60–7.71 (2H, m, Ph), 8.24 (1H, s br, NH); ^13^C-NMR (100 MHz; CDCl3) δ 16.4, 18.9, 30.9, 42.5, 55.2, 79.3, 113.9, 116.9, 117.8, 118.3, 125.3, 129.1, 129.7, 130.6, 134.8, 149.6, 155.1, 157.9, 158.8, 162.9, 169.0; HRMS calculated for C_23_H_23_NO_6_Na [M + Na]^+^: 432.1423, found: 432.1411.

### 4.4. Estimation of Minimum Inhibition Concentration (MIC) and Minimum Bactericidal Concentration (MBC)

The MIC denotes the lowest concentration of a biocidal agent (antibiotic or chemotherapeutic) that inhibits the growth of microorganisms (usually referring to bacteria and fungi), most often expressed in mg/L. It is a parameter characterizing, among others, bacteriostatic and bactericidal drugs or substances. The MIC determines what drug or compound concentration inhibits bacterial growth.

The MBC denotes the lowest concentration of bactericidal agent (antibiotic or chemotherapeutic) at which 99.9% of bacteria die. It is determined in vitro and expressed in mg/L. MBC is a parameter characterizing antibacterial drugs or compounds; it determines what concentration of the drug has bactericidal activity. A drug or compound of this concentration directly kills the vegetative forms of the bacteria.

Firstly, the 17 compounds were each diluted in *N,N*-dimethylformamide (DMF), at a final concentration of 10 mM (final concentration). Next, 50 μL of all analyzed peptidomimetics were diluted to 1 mM and plated on a 96-well plate according to the markings. MICs and MBCs were determined using a methodology described previously [28,32], with serial dilutions performed from 10^−1^ to 10^−7^. Experiments were performed with two independent replicates. An example of the analysis of test compounds of various concentrations using a resusarin dye on microplates (mg∙L^−1^) is presented in Appendix A.

### 4.5. Interaction of the Plasmid DNA with Peptidomimetics 

On the basis of MIC and MBC values, K12 and R4 strains were selected for further investigation. The bacteria were incubated with each of the 17 analyzed compounds at a concentration of 0.1 mM for 24 h at 37 °C. Next, the bacterial DNA was isolated from cultures of K12 and R4 *E. coli* using a methodology described previously [28,32].

### 4.6. Statistical Analysis.

All obtained data were presented as means ± standard error (SE). For statistical analysis, commercial Statistica packet version 5.0 was used. Parametric analyses were tested based on the Student’s *t*-test. Statistical significance of all analyzed compounds was considered at *p* < 0.05 *, *p* < 0.01 **, and *p* < 0.001 ***.

## Figures and Tables

**Figure 1 materials-13-02499-f001:**
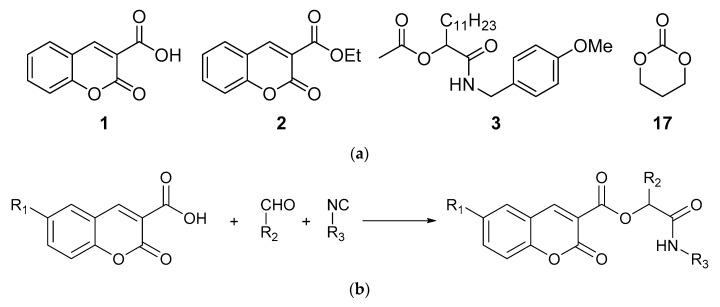
Structure of compounds (**a**) 1, 2, 3, and 17; (**b**) compounds 4–16.

**Figure 2 materials-13-02499-f002:**
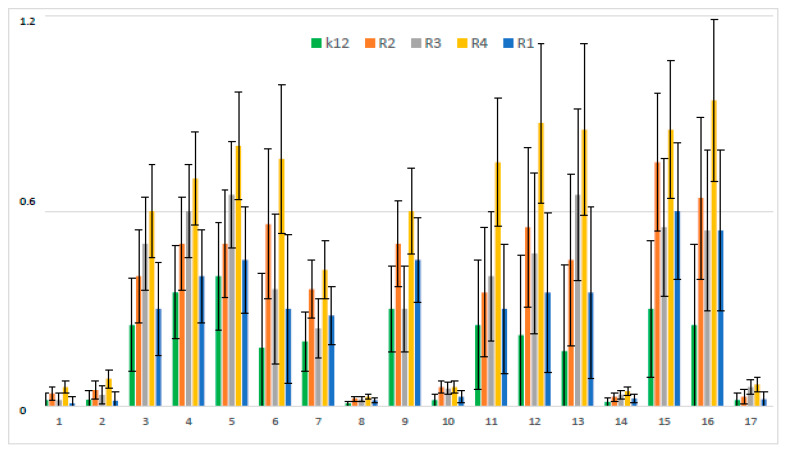
Minimum inhibitory concentration (MIC) of the peptidomimetics for investigated strains of *Escherichia coli* K12 as a control (blue), R1 and R2 strains together (red; the values were almost the same), R3 strain (gray), and R4 strain (yellow). The x-axis features coumarin derivatives 1–17 used sequentially. The y-axis features the MIC value in mM. The order in which the compounds were applied to the plate was as described in Table 1.

**Figure 3 materials-13-02499-f003:**
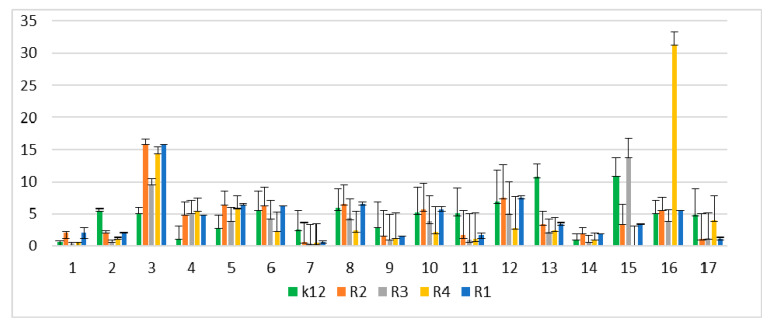
Minimum bactericidal concentration (MBC) of the peptidomimetics ratio for investigated strains of *E. coli* K12 as control (blue), R1 and R2 strains together (red; the values were almost the same), R3 strain (gray), and R4 strain (yellow). The x-axis features coumarin derivatives 1–17 used sequentially. The y-axis features the MBC value in mM. The order in which the compounds were applied to the plate was as described in Table 1.

**Figure 4 materials-13-02499-f004:**
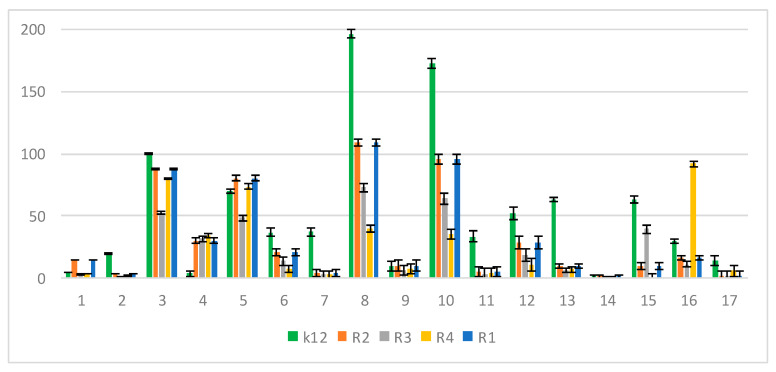
MBC/MIC of the peptidomimetics ratio for investigated strains of *E. coli* K12 as control (blue), R1 and R2 strains together (red; the values were almost the same), R3 strain (gray), and R4 strain (yellow). The x-axis features coumarin derivatives 1–17 used sequentially. The y-axis features the MBC/MIC ratio. The order in which the compounds were applied to the plate was as described in Table 1.

**Figure 5 materials-13-02499-f005:**
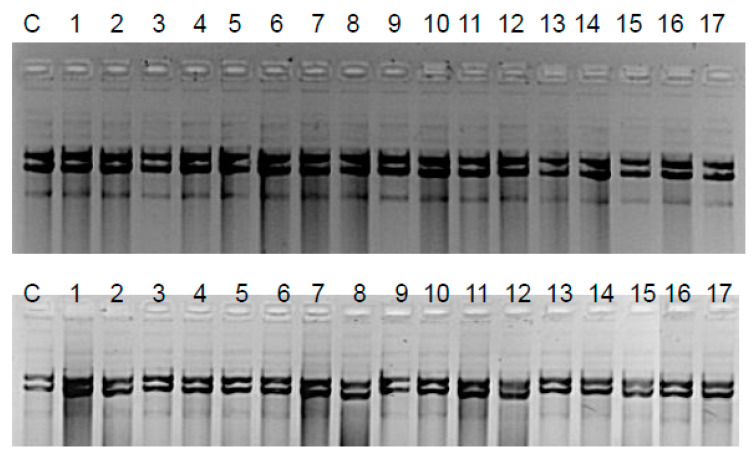
Assessment of damage to plasmid DNA isolated from K12 and R4 *E. coli* strains based on electrophoretic analysis: K12—upper gel; R4—bottom gel; C—control plasmid DNA not treated by peptidomimetics and not digested by Fpg (formamidopyrimidine DNA *N*-glycosylase/AP lyase) enzyme. The order in which the compounds were applied to the agarose gel was as described in Table 1.

**Figure 6 materials-13-02499-f006:**
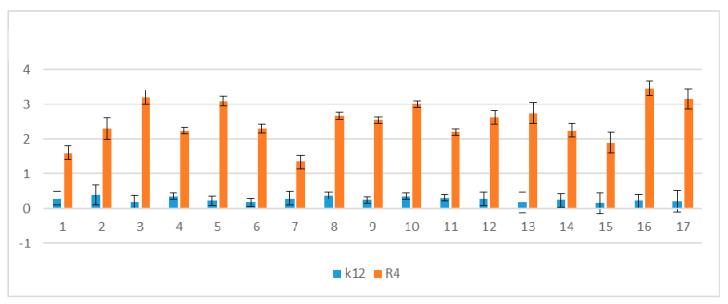
Percentage of plasmid DNA recognized by Fpg enzyme (y-axis) with control K12 and R4 strains (x-axis); C—control plasmid DNA not treated by peptidomimetics and not digested by Fpg enzyme. The compounds numbered 4–11 and 15–16 were statistically significant at <0.05 *.

**Table 1 materials-13-02499-t001:** Synthesis of compounds 4–16.

Entry	R_1_	R_2_	R_3_	Proposed Product (No.)	Present Product (No.)	Yield ^a^
1	H	C_11_H_23_	CH_2_4-OMeC_6_H_4_	4	11-Dp-606	44%
2	NO_2_	C_11_H_23_	CH_2_4-OMeC_6_H_4_	5	7-DP-595	38%
3	OMe	C_11_H_23_	CH_2_4-OMeC_6_H_4_	6	9-DP-571	17%
4	Me	C_11_H_23_	CH_2_4-OMeC_6_H_4_	7	16-DP-594	34%
5	H	C_11_H_23_	Bz	8	2-DP-577	38%
6	H	C_11_H_23_	*t*-Bu	9	10-DP-639	17%
7	H	C_11_H_23_	Cyclohexyl	10	1-DP578	29%
8	H	C_11_H_23_	CH_2_COOEt	11	6-DP-579	48%
9	H	C_7_H_15_	CH_2_4-OMeC_6_H_4_	12	5-DP-593	32%
10	H	C_5_H_11_	CH_2_4-OMeC_6_H_4_	13	4-DP-590	57%
11	H	Et	CH_2_4-OMeC_6_H_4_	14	14-DP-589	13%
12	H	Bz	CH_2_4-OMeC_6_H_4_	15	8-DP-591	72%
13	H	*i*−Pr	CH_2_4-OMeC_6_H_4_	16	17-DP-592	49%

^a^ Reaction conditions: all components of the Passerini reaction were 0.5 mmol, in dichloromethane (5 mL) for 24 h at room temperature. Proposed numbering: 1—coumaric acid, 3-DP523 present; 2—coumarin ethyl ester, 13-DP-8 present; 3–16—as outlined in the table; 17—vegan cyclone for comparison, 15-R0-93, which is an ingredient used in bodybuilding mixes for rapid muscle growth.

**Table 2 materials-13-02499-t002:** Statistical analysis of all analyzed compounds by MIC, MBC, and MBC/MIC; <0.05*, <0.01 **, <0.001***.

No. of Strain Samples	1	2	3	4	5	6	7	8	9	10	11	12	13	14	15	16	17	Type of Test
K12	-	***	-	-	***	**	-	***	-	***	**	*	*	***	**	**	***	MIC
R2	-	***	-	-	***	**	-	***	-	***	**	*	*	***	**	**	***	MIC
R3	-	***	-	-	***	**	-	***	-	***	**	*	*	***	**	**	***	MIC
R4	-	***	-	-	***	**	-	***	-	***	**	*	*	***	**	**	***	MIC
K12	-	-	**	*	**	***	-	**	-	*	-	***	-	-	-	-	-	MBC
R2	-	-	**	*	**	***	-	**	-	*	-	***	-	-	-	-	-	MBC
R3	-	-	**	*	**	***	-	**	-	*	-	***	-	-	-	-	-	MBC
R4	-	-	**	*	**	***	-	**	-	*	-	***	-	-	-	-	-	MBC
K12	-	-	-	***	-	***	*	-	**	-	**	-	**	-	**	-	-	MBC/MIC
R2	-	-	-	***	-	***	*	-	**	-	**	-	**	-	**	-	-	MBC/MIC
R3	-	-	-	***	-	***	*	-	**	-	**	-	**	-	**	-	-	MBC/MIC
R4	-	-	-	***	-	***	*	-	**	-	**	-	**	-	**	-	-	MBC/MIC

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
