# Peer review of "Coumarin Derivatives as New Toxic Compounds to Selected K12, R1–R4 *E. coli* Strains"

_materials, 2020, doi:10.3390/ma13112499_

Round 1

Reviewer 1 Report

Dear Editor, Dear Authors,

The present study shows a great interest for a worldwide medical practice problem, namely the increase of bacterial resistance to antibiotics, so the authors' concern is very current and necessary. The theme is complex, with many analyzes, with an appropriate methodology; the results are presented clearly and concisely. From my point of view, this manuscript can be published with some necessary modifications:
- the abstract and the conclusions are very long, with many details, which should be reformulated
- a closer justification of the objectives proposed and achieved in the summary and conclusions
- bibliography - must be remade
- line 30: LPS - should be explained what it is

- imunodispersive -to be corrected

- to arrange certain paragraphs according to the requirements of the journal: lines 14-40, 166-167, 240-243, 337-339, 462-472 etc.
- line: 111 - 1,3-dioxane-2-one
- line: 127 - 17 - "vegan cyclone for comparison" - What does it mean?

Author Response

- the abstract and the conclusions are very long, with many details, which should be reformulated

Yes, the abstract and conclusions have been  reformulated and shortened

(patches are marked in green). Abstract Line 14-38, Conclusions 315-339 in the new version.

- a closer justification of the objectives proposed and achieved in the summary and conclusions

Objectives :

We hypothesised that the toxicity of coumarin derivatives is dependent on the of LPS in bacteria. In order to verify this, we used K12 (smooth) and R1-R4 (rough) pathogenic E.coli strains which are characterised by differences in the type of LPS, especially in the O-antigen region, the outermost LPS layer.

-have been reformulated and corrected in line 25-27 in abstract

- bibliography - must be remade

all literature has been rewritten according to the journal's guidelines

- line 30: LPS - should be explained what it is

The abbreviation is explained on line 35 (former line 30)

- imunodispersive -to be corrected

Has been corrected

- to arrange certain paragraphs according to the requirements of the journal: lines 14-40, 166-167, 240-243, 337-339, 462-472 etc.

paragraphs and lines, 7-33, 159-160, 233-236, 330-332, 455-465 (former lines 14–40, 166–167, 240–243, 337-339, 462–472) have been ordered in accordance with the journal's requirements:

- line: 111 - 1,3-dioxane-2-one

Corrected in line 104 (former 111)

- line: 127 - 17 - "vegan cyclone for comparison" - What does it mean?

Explained in line 120 (former 127) , 17 - vegan cyclone for comparison, 15-R0-93- ingredient used in bodybuilding mixes for rapid muscle growth

Reviewer 2 Report

Paweł Kowalczyk and his colleagues (materials-797666) present an interesting investigation of the antimicrobial effects by a series of peptidomimetics with a common origin of Coumarins to the E.coli. However, I have several of the following concerns regarding the manuscript.

1. Only four strains are examined, which is not enough for proof of concept investigation. Considering the genetic diversity of the E. coli population, some strains with a clear and different genetic background is needed. Another interesting investigation would add a few more MDR E.coli. Only four strains are examined, which is not enough.

2. The pathogen in the title comes from nowhere, the available four E. coli strains, none of them has approved with the pathogenic feature, or with a clear pathogenic background. As well as I know, the K-12 strain is a lab horse strain with no evidence of pathogenic characteristics.

3. Most of the figures and tables are in bad quality and resolution, I can not read anything in the table from Line 202.

4. It would be great to list all the examined chemicals with their structure, if possible in the supplemental document, for the comparative purpose.

5. It is not clear why MBC assay is needed, a rational introduction, and sufficient discussion the results between MIC and MBC is also needed. Additionally, too many abbreviations, this should be corrected in the revised version.

minor

Technical points, 

“The MIC values revealed that the toxicity of the tested compounds increased with increasing 206 alkyl chain length, and the same effect would be expected in,”

The presentation of the manuscript should be improved

Line 40, remove "and virus"

Line 96, remove "parasite"

Line 133, which figure?

Line, only four strains are examined, which is not enough.

Lots of minor issues occurred in the current manuscript, a comprehensive proofreading and professional check is highly recommended.

Author Response

  1. Only four strains are examined, which is not enough for proof of concept investigation. Considering the genetic diversity of the E. coli population, some strains with a clear and different genetic background is needed. Another interesting investigation would add a few more MDR E.coli. Only four strains are examined, which is not enough.

Our research is a continuation of research on strains K12 and R1-R4 E. coli associated with ionic liquids (reference in manuscript number 28). Model E.coli strains for this type of research were selected because of their length of LPS major surface antigen and the virulence factor of Gram-negative bacteria which is covalently bound to the linear ECA polymer. With respect to gram-negative bacteria, smooth (smooth, S) strains synthesize LPS containing all these regions, while rough (R) strains synthesize lipooligosaccharides (LOS) composed only of lipid A substituted with a full or incomplete spinal oligosugar. Studies on the immunogenicity of some bacterial species have shown the existence of ECA immunogenic strains also among other enterobacterial species, including only rough strains, i.e. devoid of O-specific antigen and containing a full core oligosaccharide such as E. coli R1, R4 and K-12 and E. coli R2 and R3 seemed immunogenic in this type of study. It is also known from literature data that studies on the presence of anti-ECA antibodies in the serum of animals after immunization with bacterial cells showed the presence of the ECA-LPS antigen in rough strains in E. coli R1, R4, R2 and R3. Substitution of the LPS molecule with the ECA antigen alone is required for the oligosaccharide core to function fully. The ECA antigen combined with LPSs of different length in the analyzed bacterial influences seems to be particularly useful in the treatment of generalized bacterial infections, especially sepsis, caused in almost half of the cases by Gram-negative bacteria, among which species such as E. coli, Klebsiella pneumoniae or predominate P. mirabilis. The ECA antigen commonly found on the surface of bacterial cells appears to be the ideal target for specific antibodies. Opsonization of bacteria by antibodies, followed by the activation of a cascade of proteolytic reactions involving complement proteins that cause pathogen killing, is a well-known process of humoral, innate immune response.

  1. The pathogen in the title comes from nowhere, the available four E. coli strains, none of them has approved with the pathogenic feature, or with a clear pathogenic background. As well as I know, the K-12 strain is a lab horse strain with no evidence of pathogenic characteristics.

We hypothesised that the toxicity of coumarin derivatives is dependent on the of LPS in bacteria. In order to verify this, we used K12 (smooth) and R1-R4 (rough) pathogenic E.coli strains which are characterised by differences in the type of LPS, especially in the O-antigen region, the outermost LPS layer.

That is why we chose strain K12 for our research because: E.coli K-12 was originally isolated from a convalescent diphtheria patient in 1922 (Bachmann, 1972).  Because it lacks virulence characteristics, grows readily on common laboratory media, and has proven to be a valuable tool for microbial physiology and genetics research, it has become the standard bacteriological strain used in microbiological research and teaching.  E.coli K-12 is now considered an enfeebled organism as a result of being maintained in the laboratory environment forover 70 years (Williams-Smith, 1978). E. coli K-12 strains are by far the most frequently used host strains for DNA propagation (Goncalves et al., 2012, 2014). The E. coli genotype plays a significant role in host strain selection for optimal processing of cDNA, based on both quality and quantity of supercoiling (Yau et al., 2008). Several works show the potential genomic changes to improve the production of pDNA by this bacteria (Ow et al., 2009; Pablos et al., 2012; Phue et al., 2008).

E.coli K-12 has a history of safe use.  Its derivatives are currently used in a large number of industrial applications, including the production of specialty chemicals (e.g., L-aspartic, inosinic, and adenylic acids) and human drugs such as insulin and somatostatin (Dynamac, 1990).  Further, E. coli can produce a number of specialty chemicals such as enzymes which would be regulated under TSCA.  An insulin-like hormone for useas a component of cell culture media, resulting from a fermentation application in which E. coli was used as the recipient, has already been reviewed under TSCA (Premanufacture Notice P87-693).  EPA recently reviewed a submission (94-1558) for use of E. coli K-12 to produce indigo for use as a dye.  Ingeneral, E.coli K-12 is one of the most extensively studied bacteria, and has been used in genetic studies in laboratories worldwide.

Most E. coli serotypes are benign and may even contribute to normal function and nutrition in the gastrointestinal tract.  A few E. coli serotypes are pathogens. E. coli K-12 strains in use today are from standard culture collections (Bachmann, 1972), such as the American Type Culture Collection and are not recent environmental isolates.  As a result, these K-12 strains are well-characterized and should beexpected to remain as pure cultures under standard microbiological practices. 

If we use coumarin derivatives in ours as potential drugs that are the successors of antibiotics, then testing compounds on the K12 strain, which also has its industrial significance, seems justified and rational.

  1. Most of the figures and tables are in bad quality and resolution, I can not read anything in the table from Line 202.

I am giving an example of a corrected table 2

No of strain samples

1

2

3

4

5

6

7

8

9

10

11

12

13

14

15

16

17

Type of test

K12

***

***

**

***

***

**

*

*

***

**

**

***

MIC

R2

***

***

**

***

***

**

*

*

***

**

**

***

MIC

R3

***

***

**

***

***

**

*

*

***

**

**

***

MIC

R4

***

***

**

***

***

**

*

*

***

**

**

***

MIC

K12

**

*

**

***

**

*

***

MBC

R2

**

*

**

***

**

*

***

MBC

R3

**

*

**

***

**

*

***

MBC

R4

**

*

**

***

**

*

***

MBC

K12

***

***

*

**

**

**

**

MBC/MIC

R2

***

***

*

**

**

**

**

MBC/MIC

R3

***

***

*

**

**

**

**

MBC/MIC

R4

***

***

*

**

**

**

**

MBC/MIC

Table 2. Statistical analysis of all analyzed compounds by MIC, MBC and MBC/MIC test. <0,05 (5%), <0,05*, <0,01 **, <0,001***.

  1. It would be great to list all the examined chemicals with their structure, if possible in the supplemental document, for the comparative purpose.

A table with the structural formulas of the analyzed compounds was attached to the manuscript as additional materials after the references

  1. It is not clear why MBC assay is needed, a rational introduction, and sufficient discussion the results between MIC and MBC is also needed. Additionally, too many abbreviations, this should be corrected in the revised version.

The MBC test determines the lowest concentration at which an antimicrobial agent will kill a particular microorganism.  The MBC is determined using a series of steps, undertaken after a Minimum Inhibitory Concentration (MIC) test has been completed. MBC testing is useful for comparing the germ-killing activity of several antimicrobial agents at once. The MBC test can be a good and relatively inexpensive tool to rank a great number of antimicrobial agents by potency, for screening purposes.

The test parameters for the MBC are easy to control in the laboratory, so comparisons can be made fairly easily between various antimicrobial agents tested under the same conditions and their respective effects on specific microorganisms.

minor

Technical points, 

“The MIC values revealed that the toxicity of the tested compounds increased with increasing 206 alkyl chain length, and the same effect would be expected in,”

the sentence has been deleted

The presentation of the manuscript should be improved

The presentation of the entire manuscript has been corrected and improved

Line 40, remove "and virus"

the word virus was deleted on line 33 (formerly line 40)

Line 96, remove "parasite"

The word parasite was deleted on line 89 (formerly line 96)

Line 133, which figure?

Figure numer 2

Line, only four strains are examined, which is not enough.

Five strains R1-R4 and K12 were tested

Lots of minor issues occurred in the current manuscript, a comprehensive proofreading and professional check is highly recommended.

a comprehensive substantive and language correction has been made.

Round 2

Reviewer 1 Report

Dear Authors,

Dear Editor,

I read the revised manuscript and I came to the conclusion that it had been improved, and this could justify its publication.

Reviewer 2 Report

All my previous concerns were not well addressed, neither a clear explanation of rational choice of bacterial strains or the quality of figures. A previous publication does not support the work right now.

Additional comment:

Basically, my question #1 and #2 were not directly addressed.

Author Response

With reference to the earlier answers to questions 1 and 2, we just wanted to supplement them with some statements (the answers to the questions are combined)

In the present study the toxicity of coumarin and their derivatives was investigated in relation to smooth and rough Escherichia coli strains. We assumed that the microbial toxicity of the studied compounds was affected by the composition of the bacterial membrane of lipopolysaccharide (LPS) of Gram-negative bacteria. LPS is a major component of the Gram-negative, covering up to 75% of the surface. LPS is anchored in the outer membrane of Gram-negative bacteria, where it plays a crucial role not only in host-parasite interaction, but also in interaction with the surrounding environment. The LPS of E. coli comprises hydrophobic lipid A, and core oligosaccharides and polysaccharides that form the O-antigen. Rough LPS (in R-type strains) lacks the O-antigen, and has truncated core oligosaccharides.

Our hypothesis was verified using K-12 (smooth) and R1-R4 (rough) model strains which are characterised by differences in LPS, especially in the O-antigen region. In the present work, we used E. coli strains R1–R4 and E. coli K-12 and rough strains showed that differences in sensitivity to the tested compounds between smooth and rough strains of the O-antigen and truncated oligosaccharide core which may play important roles in the cellular response to coumarin derivatives. Membrane rearrangements and disruption may in turn result in changes of bacterial responses to other biologically active compounds such as antibiotics. However, the toxicity of coumarin derivatives is strongly correlated with the strains due to differences in the structure of LPS. It appears the rough strains of E. coli were affected by coumarin derivatives to a greater extent  than the smooth K-12 strain, which has a full-length O-antigen LPS structure. By analogy to ionic liquids we have conducted similar studies with the use of coumarin derivatives. Strain selection was based on our previous research [1,2]. Bacteria treated by coumarin derivatives concerning with a specific group of carbon atoms carbons in their alkyl chains and containing in its construction carboxylic acid, hydroxyl groups or esters had positive antimicrobial effects which was observed in previous studies with ionic liquids [1,2]. In our previous studies, quaternary alkylammonium bromides (precursors) as well as theophylline-based alkylammonium ionic liquids (TILs) were investigated, and ILs containing theophylline as an anion displayed efficient feeding deterrence and antifungal activity, as well as potential antibacterial activity [1,2]. ILs usually include an alkyl chain that may influence amphiphilic properties, and the presence of a hydrophobic region may result in strong interactions with lipid membranes.

This is especially important in the case of Gram-negative bacteria surrounded by an outer membrane containing lipopolysaccharide (LPS), since components of the bacterial outer membrane potentially affect interactions with ILs, thereby influencing the toxicity of ILs as well as differences in peroxidase activity and lipid peroxidation.

In relation to previous studies with ionic liquids, similarly after using coumarin derivatives we have observed that the length of the alkyl chain can strongly affect the toxicity of ILs, and a characteristic ‘cut-off’ effect has been noted, since the correlation between increasing alkyl chain length and toxicity ceases for longer alkyl chains.

We showed, for the first time, that coumarin derivatives can strongly interfere with bacterial membranes in relation to previous studies with ionic liquids which also causing their rapid changes of the zeta potential and membrane lateral diffusion [1,2].

[1] Borkowski A., Kowalczyk P., Czerwonka G., Cieśla J., Cłapa T., Misiewicz A., Szala M., Drabik M. (2017) Interaction of quaternary ammonium ionic liquids with bacterial membranes - Studies with Escherichia coli R1–R4-type lipopolysaccharides. Journal of Molecular Liquids 246, 282–289.

[2] Paweł Kowalczyk, Andrzej Borkowski, Grzegorz Czerwonka, Tomasz Cłapa, Jolanta Cieśla, Anna Misiewicz, Marta Borowiec, Mateusz Szala. The microbial toxicity of quaternary ammonium ionic liquids is dependent on the type of lipopolysaccharide. Journal of Molecular Liquids 266 (2018) 540–547.

Round 3

Reviewer 2 Report

There are major critics remaining.

  1. To be precise, remove the word "pathogenic", in the title as well as the whole context. Actually, You provide "zero" evidence that all these are pathogenic to what, humans, any type of animals?
  2. Figure 2 is too raw, and the quality is bad, I suggest to put it as a supplemental figure. 
  3. Figure 3/4/5 are too busy to visualize, considering the huge standard deviation bar, it is even hard to look in figure 3. What is (0,2  0,4 ...) in y-axis of figure 2 means? Is it correct? Remove the "-50" in Figure 5. Is there a way to improve these figures?

Author Response

All reviewers comments have been taken into account:

  1. To be precise, remove the word "pathogenic", in the title as well as the whole context.

The word „pathogenic” has been removed from both the title and the entire manuscript

Actually, You provide "zero" evidence that all these are pathogenic to what, humans, any type   of animals?

       Because bacteria can induce sepsis, I meant pathogenicity to humans. I am sorry that I did not write this issue more precisely.

  1. Figure 2 is too raw, and the quality is bad, I suggest to put it as a supplemental figure.

 Thank you very much for the suggestions, figure 2 is already added in supplementary materials

  1. Figure 3/4/5 are too busy to visualize, considering the huge standard deviation bar, it is even hard to look in figure 3.

 Figure 3. Minimum inhibition concentration (MIC) of the peptidomimetics for investigated strains of E. coli blue colour K12 as control, red R1 and R2 strain together (the values were almost the same), grey R3 strain, yellow-R4 strain. Abscissa axis (X)- numbers 1-17 coumarin derivatives used sequentially. Ordinate axis (Y) MIC value in mM. The order in which the compounds were applied to the plate was as follows in Table 1.

Figure 3 has been corrected (stretched horizontally). The scale has been changed to better read the drawing. The values on the X and Y scales have been signed and marked in yellow below the drawing.

  1. What is (0,2  0,4 ...) in y-axis of figure 2 means? Is it correct?

Values 0.2..0.4 mean MIC value in mM.  Yes it is correct

  1. Remove the "-50" in Figure 5. Is there a way to improve these figures?

The value of -50 from Figure 5 has been removed. I can only increase the scale for numerical values.

Due to the inclusion of Figure 2 in additional materials, the numbering under the drawings changed in the main text. At present drawing 2 is the former drawing 3, etc. and are marked in yellow in the text.
